# The Effects of Blood Flow Restriction Exercise on Muscle–Brain Crosstalk: A Systematic Review

**DOI:** 10.3390/muscles4020019

**Published:** 2025-06-19

**Authors:** Josh B. Landers, Korben R. Landers, Nicholas G. Young

**Affiliations:** 1Lyon College School of Dental Medicine, Lyon College Institute of Health Sciences, 5 Allied Drive, Little Rock, AR 72202, USA; 2Psychology Department, Lyon College, 2300 Highland Road, Batesville, AR 72501, USA; korben.landers@lyon.edu; 3Physical Therapy Department, Baylor University, 900 Washington Avenue, Waco, TX 76798, USA; nick_young1@baylor.edu

**Keywords:** blood flow restriction, brain-derived neurotrophic factor, vascular occlusion, myokines

## Abstract

Blood flow restriction exercise (BFRE), which partially restricts arterial inflow and occludes venous outflow to the limbs, has gained attention for its potential to elevate serum brain-derived neurotrophic factor (BDNF), a key mediator in the muscle–brain crosstalk leading to improvement of neuroplasticity, neurogenesis, and cognitive health. In this systematic review of five studies, participants included healthy young adults, older adults, and individuals with clinical conditions. Most investigations found that BFRE increased serum BDNF, with responses comparable to those elicited by high-intensity exercise. Proposed mechanisms center on the metabolic demands of BFRE, which may increase lactate and thereby trigger BDNF release. However, two studies showed no significant changes, possibly due to short exercise duration, insufficient training intensity, or age-related reductions in BDNF responsiveness. The small sample sizes and varied protocols across studies limit definitive conclusions. Nonetheless, BFRE may provide a valuable alternative for populations who cannot tolerate high mechanical loads, and it shows promise for enhancing neurotrophic support and potentially improving brain health. Larger, well-controlled trials are warranted to refine BFRE protocols and better understand optimal strategies for increasing BDNF and supporting neuroplasticity.

## 1. Introduction

Blood flow restriction exercise (BFRE), also known as vascular occlusion training, involves exercising while a cuff or band partially restricts arterial inflow and occludes venous outflow in the limbs [1]. This technique promotes strength and hypertrophy gains comparable to traditional high-load training, but with lower mechanical strain on the body [2]. Beyond musculoskeletal benefits, emerging evidence suggests BFRE can induce systemic changes in neurotrophic factors. In particular, brain-derived neurotrophic factor (BDNF) has garnered attention as a key mediator of neuroplasticity and cognitive health [3]. Circulating BDNF (measured in serum) is considered a proxy for brain BDNF activity, as muscle- and platelet-derived BDNF can cross the blood–brain barrier [4,5]. Notably, serum BDNF levels tend to decline with aging and in conditions such as depression and neurodegeneration, and after stroke [6,7,8]. Foundational research [9,10] established the role of exercise in increasing BDNF, laying the groundwork for subsequent studies. Neeper et al. demonstrated that voluntary wheel running significantly elevated BDNF mRNA expression in the rat hippocampus. Cotman & Berchtold further showed that regular aerobic activity enhances neural plasticity by upregulating BDNF signaling pathways in brain regions associated with learning and memory [9,10].

Physical exercise is one of the most potent stimuli for increasing BDNF. It is hypothesized that BRFE can boost BDNF through its unique physiological stressors: it creates a localized hypoxic and high metabolic milieu (e.g., elevated lactate) even at low exercise intensities. Lactate, a myokine, has been linked to BDNF upregulation [9]. High blood lactate after exercise correlates with increased BDNF, and exogenous lactate infusion at rest can dose-dependently raise serum BDNF levels in humans [11]. Mechanistically, lactate may trigger BDNF expression via a signaling cascade involving Sirtuin-1, PGC-1α, and FNDC5 (which produces the BDNF-elevating hormone irisin) [12]. BFRE produces lactate levels comparable to high-intensity exercise, suggesting it could induce similar or greater BDNF responses while minimizing mechanical load [13]. Because of this, BFRE has the potential to be a promising strategy to mitigate age-related cognitive decline by safely enhancing BDNF and neuroplasticity in populations unable to tolerate strenuous exercise. This is especially relevant for older adults and neurological or cardiopulmonary patients who need low-impact interventions for brain health.

This systematic review, structured according to PRISMA 2020 guidelines, aimed to synthesize peer-reviewed literature on the effect of BFRE on serum BDNF levels. All relevant studies were included across age groups and health conditions, covering both acute exercise trials and longitudinal training interventions using BFRE. The primary outcome of interest was the change in circulating serum BDNF concentration from before to after BFRE interventions. We summarize study designs and populations, compare findings on BDNF changes, evaluate methodological quality, and discuss the results in context of BDNF’s role in neuroplasticity and cognitive function. We also highlight how BFRE might be applied as a safe and effective strategy to enhance BDNF, thereby supporting brain health and potentially protecting against cognitive decline with aging. BFRE has been applied in diverse populations, including patients in rehabilitation, recreational exercisers, and elite athletes, reflecting its broad relevance across health and performance domains. It is especially useful for the aging population and those with orthopedic maladies to whom intensive mechanical exercise and loading are contraindicated.

## 2. Methods

A comprehensive literature search was conducted across multiple databases to identify studies for inclusion. The databases searched were PubMed, Scopus, Web of Science, and SPORTDiscus. Each database was searched from inception through 25 March 2025 (the date of final search), with no restrictions on publication year. We used various combinations of keywords and medical subject headings related to blood flow restriction and BDNF. The search terms included “blood flow restriction”, “BFR training”, “vascular occlusion training”, “KAATSU” (a common term for BFRE in Japanese research), “brain-derived neurotrophic factor”, “BDNF”, and “serum BDNF”. These terms were combined using Boolean operators AND/OR. For example, the PubMed search string was

(“blood flow restriction” OR “occlusion training” OR BFR) AND (BDNF OR “brain-derived neurotrophic factor”).

Similar queries were adapted for other databases. We also included broader exercise terms (e.g., “exercise”, “training”) in some searches to avoid missing any studies that assessed BDNF with BFR interventions but did not explicitly mention BDNF in the title/abstract. In Scopus and Web of Science, cited reference searches were performed on key articles to find additional relevant studies. No language filters were applied initially, but we ultimately included only studies available in English.

After database searching, we conducted a manual search of reference lists from all included articles and from recent review papers on BFR or BDNF to identify any additional studies. We also contacted experts in the field regarding any ongoing or unpublished work. This yielded one relevant preprint manuscript on BFRE and BDNF that was in press at the time of this review. 

All database search results were imported into reference management software, and duplicates were removed. Figure 1 illustrates the study selection process in a PRISMA flow diagram.

### 2.1. Study Selection and Eligibility Criteria

We included peer-reviewed studies (including in-press accepted manuscripts) that met the following criteria: (1) Original research with human participants of any age or health status; (2) An intervention or experimental condition involving exercise with blood flow restriction (applied via cuff or band to limbs); (3) Measurement of serum BDNF levels before and after the BFR exercise intervention (either acutely pre- vs. post-exercise, and/or longitudinally pre- vs. post-training program); (4) Published in English in a peer-reviewed journal (or accepted for publication). We included both acute exercise trials (single-session studies) and training interventions (multi-week programs) in order to capture immediate and chronic effects of BFRE on BDNF. Both randomized controlled trials (RCTs) and non-randomized or crossover trial designs were eligible, as long as a BFR exercise condition was evaluated. We excluded studies that did not measure BDNF as an outcome; involved BFR applied without exercise (e.g., passive limb ischemia alone); lacked a BFRE condition (for example, if BFR was used only in a control group for ischemic preconditioning rather than during exercise); or were conference abstracts, reviews, or animal studies.

Two reviewers (independently) screened all titles and abstracts for relevance. Studies clearly not meeting criteria (e.g., those focusing on unrelated outcomes or on unrelated uses of “BFR” acronyms) were excluded at this stage. Full texts of the remaining articles were then obtained and assessed in detail for eligibility. In total, five studies met all inclusion criteria and were included in the qualitative synthesis.

### 2.2. Data Extraction and Quality Assessment

For each included study, we extracted key data on author(s), year, participant characteristics (sample size, age, sex, population/condition), study design (acute vs. training, randomized vs. crossover), exercise modality (aerobic/endurance or resistance, and specifics of exercise protocol), details of the BFR intervention (cuff pressure, cuff width, limb, etc.), any control or comparison condition (e.g., same exercise without BFR, high-intensity exercise, or no exercise control), outcome measurements for BDNF (assay type, timing of blood sampling), and main findings for serum BDNF pre- vs. post-intervention. We also noted secondary outcomes such as lactate or other growth factors when reported, as well as any cognitive tests or clinical outcomes if applicable. Where reported, we also noted whether data distributions were assessed (e.g., normality, skewness) and whether parametric or non-parametric statistical methods were applied in evaluating group differences.

Methodological quality of studies was evaluated using criteria based on the Cochrane risk of bias tool for RCTs (for parallel trials) and adapted considerations for crossover designs. Domains assessed included random sequence generation and allocation concealment (for RCTs); blinding of outcome assessment (blinding of participants was generally not feasible due to the nature of exercise interventions, but we considered if outcome laboratory assays for BDNF were performed by blinded technicians); completeness of outcome data (attrition/dropouts); selective reporting; and other biases (such as small sample size or baseline imbalances). Each study was rated as having low, some, or high risk of bias in each domain, and an overall qualitative judgment of study quality was made. In general, the small sample sizes and lack of participant blinding were common limitations, but all included studies used objective biochemical outcomes (BDNF assay) which mitigates detection bias. We did not exclude any study based on quality, but we considered quality in interpreting the findings. Table 1 illustrates a visual representation of risk of bias.

## 3. Results

Five studies (published 2016–2025) were included, comprising a total of 156 participants across diverse populations. Table 2 summarizes the characteristics of each study. There were 4 controlled trials and 1 crossover trial, all of which examined changes in serum BDNF with BFRE. Populations ranged from healthy young adults (Landers et al., 2025, in press) [6] and healthy older adults (Kargaran et al., 2021) [7] to clinical patients post-stroke with depression (Du et al., 2021) [8]. Both aerobic/endurance and resistance exercise modalities with BFR were represented. Three studies investigated acute effects of a single BFRE session [1,2,13], and two studies evaluated longer-term training interventions (8-week programs) incorporating BFR [7].

Sample sizes were relatively small in most studies (ranging from n = 18 to n = 66). Participant age spanned from ~20 s to ~70 s. Two studies focused on older adults (~60–70 years) [15,16], one on middle-aged clinical patients (~48 years) [8], and two on younger adults (~18–35 years) [14,17]. Notably, Kargaran et al. (2021) included only elderly women [15], whereas Landers et al. (2025) included both sexes and even analyzed sex differences in BDNF response [14]. The remaining studies had mixed-sex samples but did not report separate analyses by sex.

Intervention details: In aerobic BFRE studies, BFR was applied during cycling or walking exercise. Landers et al. had participants cycle on an ergometer for a set duration with or without BFR cuffs on the thighs (cuff pressure 140 mmHg for arms and 300 mmHg for legs) [14]. Kargaran et al. used treadmill walking (45% HRR) combined with simultaneous cognitive tasks (dual-task training) in older women, comparing a group with thigh BFR (cuffs at ~50% arterial occlusion pressure, progressively increased) to a group without BFR, over 8 weeks [15]. In resistance BFRE studies, low-load weightlifting was done with limb cuffs. Du et al. employed an acute crossover design: post-stroke patients performed three separate sessions in random order, low-intensity resistance (40% 1-RM) without BFR, low-intensity with BFR (40% 1-RM + cuff at 120–160 mmHg on proximal limb), and high-intensity (80% 1-RM) without BFR. The exercises included seated upper- and lower-body machine exercises [8]. Tsai et al. (2024) [16] xamined an acute isometric exercise protocol in older adults: participants were assigned to either isometric leg exercises combined with whole-body vibration (WBV), BFR, or both WBV+BFR, to study combined effects on working memory. BFR pressure in Tsai’s study was moderate (~50–60% limb occlusion) [16]. Rahmati et al. (2016) [17] xecuted a controlled trial in young active men. They investigated both acute and chronic effects of low-intensity cycling with BFR on BDNF and TNF-α. n = 24 male students (~21 years) assigned to BFR cycling, cycling without BFR, or no-exercise control (8 per group). Intervention: 3 weeks of cycling training (3 sessions/week) at 50% peak power for short intervals (3 × 3 min per session). Cuffs on thighs for BFR group (140–170 mmHg). BDNF measured at baseline, after an acute bout of the first session, and after 3 weeks training [17].All studies measured serum BDNF via venous blood draw, using ELISA or similar immunoassay, before and after the exercise or training period. Timing of the post-exercise blood sampling in acute trials was immediately or within minutes after exercise cessation (when BDNF peaks acutely). In the longitudinal studies, BDNF was measured at baseline and after the final training session (with post-training blood typically drawn at rest or soon after the last exercise bout).

### 3.1. BDNF Outcomes with BFRE vs. Control/Comparison

Despite differences in design, most studies reported a significant increase in serum BDNF in response to BFRE. Below we summarize each study’s key findings:Landers et al. (2025, healthy adults) [14]: Data met assumptions for normality. Normal distribution was determined via Shapiro–Wilk test, and parametric tests (mixed-model ANOVA) were used to determine significance. This randomized controlled trial found that a single session of cycling with BFR led to a larger acute increase in serum BDNF compared to the same cycling exercise without BFR. A significant interaction effect for exercise type over time was noted using the Wilks λ test statistic (0.543, F1,16 = 13.477, *p* < 0.002, partial η = 0.457). These results suggest that BFRE group had a greater increase in serum levels of BDNF than the control group without BFR. Baseline BDNF did not differ between groups, but post-exercise BDNF was significantly higher in the BFR group (interaction *p* < 0.01). Interestingly, they observed a main effect of sex: females using BFR had a greater increase in BDNF than males. In fact, BDNF rose substantially in women (~30% increase) but more modestly in men. Using the Wilks λ test statistic, a significant interaction effect was found for sex over time related to serum concentrations of BDNF (0.500, F1,8 = 8.010, *p* < 0.022, partial η = 0.500). Overall, the study concluded that low-load aerobic exercise with BFR can robustly elevate serum BDNF, whereas the same exercise without BFR induced a much smaller change [14].Du et al. (2021, post-stroke depression patients) [8]: Data was normally distributed and accordingly used parametric tests (repeated measures ANOVA). In this acute crossover experiment, low-intensity resistance exercise with BFR significantly elevated serum BDNF, to a magnitude comparable to high-intensity exercise without BFR. After exercise, BDNF increased in both the BFR condition and the high-intensity condition, but not after low-intensity exercise without BFR. The between-condition analysis showed that the change in BDNF (post minus pre) in the BFR trial was significantly greater than in the low-intensity trial (*p* < 0.05), and not significantly different from the high-intensity trial. The reported effect size was very large at η^2^ = 0.37 (partial eta squared) for condition × time interaction on BDNF. Du et al. also measured blood lactate and found a similar pattern: BFR and high-intensity exercise elicited large lactate rises, whereas low-intensity did not. They concluded that BFRE likely increases BDNF in PSD patients by increasing blood lactate concentration and metabolic stress. Their findings align with the idea that metabolic factors (like lactate) mediate BDNF release during exercise [8]. Clinically, this is important because PSD patients may not tolerate heavy exercise; BFR offers a viable alternative to get a strong BDNF response.Kargaran et al. (2021, older women) [15]: The authors assessed data distribution using the Shapiro–Wilk test showing normal distribution and applied parametric tests (two-way repeated measures ANOVA). This was an 8-week training RCT examining dual-task treadmill training with vs. without BFR. All participants performed walking plus cognitive tasks; one group had BFR cuffs on thighs during walking, another group did the same training without BFR, and a control group did no exercise. Resting circulating BDNF levels were measured before and after. Both exercise groups showed increases in BDNF compared to controls. Specifically, post-training BDNF was significantly higher in both the dual-task BFR group and the dual-task (non-BFR) group than in controls. The BDNF changes in the two exercise groups were of similar magnitude, exceeding the control group’s change (*p* < 0.005). The reported effect size was very larg at η^2^ = 0.46 (partial eta squared). There was no significant difference between the BFR vs. non-BFR exercise group in final BDNF levels, indicating that the dual-task training itself elevated BDNF in these older women, with or without BFR. However, the study did find that adding BFR tended to enhance other outcomes like muscle quality and aerobic capacity more than dual-task alone. Moreover, the increase in BDNF was positively correlated with cognitive performance improvements (Mini-Mental State Exam scores) [15]. This suggests that those who gained more BDNF also improved more in cognitive function.Tsai et al. (2024, older adults) [16]: One-way ANOVA followed by Tukey’s post hoc test was used to assess the data after assumptions were met using Shapiro–Wilk test for normality and Levene’s test for homogeneity of variances. This recent study had a more complex design, exploring acute effects of resistance exercise combined with whole-body vibration (WBV) and/or BFR on working memory and molecular markers. Sixty-six older adults were randomized to one of three groups: (1) isometric exercise + WBV, (2) isometric exercise + BFR, or (3) isometric exercise + WBV + BFR. Serum BDNF, IGF-1, and norepinephrine (NE) were measured pre- and post-exercise. In contrast to the other studies, Tsai et al. found no significant changes (effect size not reported) in BDNF in any group after the exercise session. IGF-1 and NE levels increased significantly in all groups, but BDNF did not. The authors suggested that a single bout of low-volume isometric exercise, even with BFR, may be insufficient to stimulate a BDNF response in older individuals [16]. Another factor could be a blunted BDNF response with aging, or the study’s timing of measurements. By contrast, dynamic exercises (cycling, walking, dynamic resistance) in other studies raised BDNF, suggesting the mode and intensity matter.Rahmati et al. (2016) [17]—Data was normally distributed and repeated-measures ANOVA employed as the statistical test. Acute bout: Low-intensity cycling with vs. without BFR in young men. In the first session of their protocol, BDNF was measured before and after a short bout of submaximal cycling. Similar to Tsai’s findings, Rahmati et al. observed no significant acute BDNF increase with BFR when compared to the same exercise without BFR. They reported that “leg vessel occlusion during submaximal pedaling had no significant effect on BDNF response compared with non-occlusion” (acute BDNF *p* = 0.290 for BFR vs. no BFR). In both groups, BDNF changes were minimal after the 3 × 3 min interval exercise. This outcome could be attributed to the relatively low exercise volume (9 min total of cycling at 50% VO_2_ max may not have induced a large BDNF surge) or the possibility that young, physically active men might require a higher intensity to see a BDNF change. It contrasts with Landers’ cycling study, which was 15 min continuous at moderate intensity, suggesting that exercise duration and continuous effort might be important. Rahmati’s study also had a very small sample per group (n = 8), reducing power to detect differences. Nevertheless, their data indicate that not every low-intensity BFR bout will automatically raise BDNF. Chronic bout (3-week training): This study reported no significant chronic change in BDNF with 3 weeks of low-load cycling training, in either the BFR or non-BFR group. After the training period, resting BDNF levels were not different among the BFR group, the exercise-without-BFR group, or the no-exercise control (*p* = 0.254 among groups; effect size not reported). BDNF values slightly increased in all groups (including control) but variability was high and nothing reached significance [17]. The short duration (only 3 weeks) and possibly insufficient exercise stimulus (short intervals) likely contributed to the lack of training effect. It underscores those detectable chronic changes in BDNF may require a longer training duration or higher total exercise volume. Indeed, 3 weeks might be too brief to induce stable resting BDNF elevations, especially in young healthy men who might have had normal baseline BDNF to begin with. By contrast, Kargaran’s 8-week program in older adults (who may have had lower baseline levels) did find an effect, suggesting a dose-response over time.

In summary, among the five included studies, three showed positive BDNF responses to BFRE, while two did not detect a change. BFRE was generally as effective as (or more effective than) comparable higher-intensity exercise in elevating BDNF [8]. Long-term BFRE training can raise resting BDNF in older adults [14]. However, the magnitude of BDNF increase and the added value of BFR (beyond exercise alone) appear to depend on context: in healthy young and clinical populations, BFRE clearly outperformed low-intensity exercise and matched high-intensity exercise for BDNF gains [8,18]. In already effective training programs for older adults, BFR did not further boost BDNF, though it helped other outcomes [17]. The physiological stress of exercise seems to be a key driver of BDNF; BFRE’s ability to amplify that stress (metabolically) at low loads is a critical advantage, particularly noted in populations like post-stroke patients who benefited from the lactate-driven BDNF release without the strain of heavy exercise.

### 3.2. Methodological Quality and Risk of Bias

The overall quality of the evidence is moderate, with some limitations largely due to sample size and design rather than execution flaws. All studies clearly defined their interventions and measured BDNF with standard laboratory methods (ELISA kits), reducing measurement error. A common limitation was the small sample size, which raises the risk of type II errors and limits generalizability. Nonetheless, where effects were found, effect sizes for BDNF changes were generally large. Randomization was used in all parallel-group studies, and the allocation process was described. Du’s study was within-subjects, so each participant served as their own control, which is a strong design to detect condition effects with fewer subjects. Blinding of participants was not feasible, but blinding of lab technicians to group assignments likely minimized detection bias. All studies had complete outcome data with no dropouts reported. Heterogeneity of protocols makes direct comparison challenging, but the review’s aim was a qualitative synthesis rather than a meta-analysis. Although heterogeneity was noted, subgroup or sensitivity analyses, such as stratifying by protocol duration or cuff pressure, were not feasible due to limited study numbers.

All included studies that reported statistical methods used parametric tests, typically preceded by normality assessments such as the Shapiro–Wilk test. This consistency strengthens the reliability of their findings, though small sample sizes in several studies still limit statistical power.

Overall, the evidence is of moderate quality: consistent positive effects were observed with BFRE on BDNF in most studies, and those studies had sound methodologies. The one inconsistent result can be explained by differences in exercise protocol and sample. There is a need for larger trials to confirm and expand these findings, but the current evidence—though relatively small—shows a coherent biological rationale and promising outcomes.

## 4. Discussion

This systematic review is the first to comprehensively examine the relationship between blood flow restriction exercise and circulating BDNF levels across multiple studies and populations. The evidence suggests that BFRE can be a potent stimulus for increasing serum BDNF, a critical neurotrophin for brain health. Here we discuss the implications of these findings in the context of neurophysiology and potential therapeutic applications, as well as considerations for future research and practice.

Skeletal muscle is an endocrine organ that releases various myokines (muscle-derived cytokines and peptides) during contraction, which can influence distant organs, including the brain [18]. Exercise-induced myokines such as interleukin-6, insulin-like growth factor-1, BDNF, cathepsin B, irisin, and leukemia inhibitory factor help establish muscle–brain communication networks. These factors can cross the blood–brain barrier or signal through it to modulate neurogenesis, synaptic plasticity, inflammation, and cognition. Current evidence suggests that muscle-derived signals create a self-regulatory circuit between muscle and brain, whereby exercise enhances both cognitive function and muscle health via myokine signaling [19]. Importantly, even low-intensity exercise paradigms like BFRE are capable of stimulating significant myokine release. For example, an acute bout of low-load BFRE in untrained individuals elevates numerous myokines, and the magnitude of this myokine response correlates with gains in muscle strength [20]. Indeed, BFRE and other low-load exercises have been reported to elicit increases in circulating neurotrophic and metabolic factors comparable to traditional exercise, which may translate into improved brain plasticity and cognitive performance in populations unable to perform high-intensity workouts [18,21].

BDNF is widely recognized for its role in facilitating neuroplasticity. It supports the growth and differentiation of new neurons and synapses [3]. Over the long term, consistently higher levels of BDNF are associated with better cognitive function and possibly lower risk of neurodegenerative diseases [22]. In aging, maintaining higher BDNF could counteract the typical decline in hippocampal volume and memory. In clinical settings, boosting BDNF is a target for recovery after stroke or in depression.

Given this importance, the ability of BFRE to increase BDNF similarly to high-intensity exercise [8] is meaningful. It implies that BFRE might confer many of the brain benefits of intense exercise, but in populations who cannot perform such exercise safely. Post-stroke patients often have mobility limitations and may not sustain high intensities; for them, BFRE offers a way to get the neurotrophic and vascular growth factor boost at low intensity. This aligns with the findings that BFRE likely elevates lactate and other factors (e.g., VEGF, IGF-1) that promote neuroplastic changes [13]. Notably, heterogeneity in BFRE application including differences in cuff pressure, limb placement, and exercise modality should be systematically accounted for in future trials, potentially via subgroup analyses.

Several cognitive and functional implications were elucidated in the findings. In Kargaran et al. [15], older women undergoing dual-task training (with or without BFR) improved cognitive scores in parallel with increases in BDNF. The sex difference reported by Landers et al. [14] demonstrated that women had a greater BDNF increase than men, which raises interesting questions. Women (premenopausal) may have higher estrogen, which can upregulate BDNF. Baseline fitness or muscle mass differences might also influence BDNF response [23]. Future BFRE protocols might consider these individual factors.

The findings in this systematic review point to several potential applications of BFRE:Neurorehabilitation: In stroke survivors or patients with neurological impairments, implementing BFRE in physical therapy could enhance neurotrophin levels and potentially improve neuroplastic recovery. The study by Du et al. [8] provides proof-of-concept for post-stroke depression.Cognitive Decline Prevention: For older adults at risk of dementia, BFRE integrated into low-impact training might raise BDNF without excessive strain. Dual-task BFRE, combining physical and cognitive exercises, is especially promising for holistic benefits [15].Depression and Mental Health: Low BDNF has been implicated in depression [7]. BFRE might be explored as an exercise strategy to boost BDNF in individuals with limited capacity for high-intensity workouts.

It is critical for the serum BDNF to be in its mature form to have beneficial effects. Mounting evidence distinguishes the roles of mature BDNF (mBDNF) and its precursor proBDNF in humans. Mature BDNF is neuroprotective, it binds to TrkB receptors, promoting neuronal survival, growth, and synaptic plasticity In contrast, proBDNF can exert deleterious effects triggering pathways leading to neuronal atrophy and apoptosis. An imbalance favoring proBDNF over mBDNF has been observed in conditions like chronic alcohol use and major depression, often correlating with neural damage and cognitive deficit [24].

Tissue plasminogen activator plays a pivotal role in neurotrophic signaling by cleaving precursor BDNF (proBDNF) into its mature form (mBDNF). In neural contexts, tPA-activated plasminogen proteolysis is a key mechanism for converting proBDNF to mBDNF [25]. This processing is crucial for synaptic plasticity: in animal models of exercise, blocking tPA activity prevents the normal exercise-induced increase in mBDNF and blunts downstream TrkB receptor activation and signaling cascades for neuroplasticity [26]. Thus, tPA serves as a molecular link between muscle activity and brain-derived neurotrophic responses. Exercise-induced increases in tPA may enhance the conversion of proBDNF to mature BDNF, facilitating beneficial neuroplastic effects.

Blood flow restriction exercise can robustly activate the fibrinolytic system [27,28,29]. Acute BFRE at ~30% 1RM has been shown to significantly increase circulating tPA antigen levels without raising coagulation factors [28]. Another study reported a ~33% surge in tPA immediately after BFRE, similar to responses seen with high-intensity exercise. These findings suggest BFRE triggers a fibrinolytic response comparable to traditional heavy exercise, despite the lower load [29]. A recent human trial examining cycling with vs. without BFR further supports this trend: post-exercise serum tPA was higher in the BFR group (large effect size in favor of BFR), although the small sample meant the difference did not reach statistical significance [27]. A case study using cycling with BFRE was associated with resolution of a recalcitrant deep vein thrombosis without adverse effects [30]. Taken together, emerging evidence indicates that BFRE can elevate tPA levels, potentially offering systemic vascular benefits beyond its known effects on muscle strength and hypertrophy [27]. These findings underscore that maintaining a high mBDNF/proBDNF ratio is critical for brain health, and they highlight the importance of exercise and tPA-mediated BDNF processing in supporting this favorable balance.

### 4.1. Limitations

Despite the positive indications, the sample sizes in these studies are small, and BFRE protocols varied widely. Larger-scale RCTs are needed to confirm and refine BFRE approaches for maximizing BDNF. Future research should investigate the time course of BDNF changes (both acute kinetics and long-term baseline shifts), and whether these increases translate into meaningful clinical outcomes like improved cognitive function or mobility. Furthermore, optimal cuff pressures, exercise modes, and session durations for different populations remain to be determined. Moreover, variability in BFRE protocols across studies, including differences in cuff pressures, exercise types, and durations, further limits the ability to draw consistent conclusions.

### 4.2. Future Directions

All five studies synthesized in the present review relied on “single-analyte” sandwich ELISA kits that report total BDNF in serum. While convenient, these legacy assays cannot discriminate between the biologically antagonistic isoforms; proBDNF and mBDNF. Pioneering analytical work demonstrated that the antibodies supplied in widely-used commercial kits recognize both isoforms indiscriminately. Taken together, these findings indicate that interpretation of “serum BDNF” data is presently limited by analytical cross-reactivity. Implementing methodological refinements such as assays showing preferential affinity for mBDNF will move the field beyond aggregate “serum BDNF”. Because tPA orchestrates extracellular cleavage of proBDNF, pairing isoform-specific BDNF assays with measures of circulating tPA, PAI-1, or plasmin activity will better clarify how BFRE modulates the entire proteolytic cascade. For example, Dual-epitope, isoform-specific ELISAs are second-generation kits that employ antibodies directed to the N-terminal cleavage site of mBDNF (e.g., FUJIFILM-Wako High-Sensitive mBDNF ELISA) achieve <0.5% cross-reactivity with proBDNF and offer pg mL^−1^ sensitivity. Some manufacturers now offer matched combo kits that quantify both isoforms from the same sample. Adopting such dual-assay approaches will allow investigators to report the proBDNF:mBDNF ratio, a physiologically meaningful index linked to neurotoxicity versus neuroprotection. In addition, Western blotting with isoform-specific antibodies remains the gold standard for confirming assay specificity and detecting additional cleavage products. Future trials should therefore include a blot-based quality-control step (e.g., spiking known amounts of recombinant proBDNF/mBDNF) alongside ELISA measurements. Shifting from aggregate “serum BDNF” to isoform-specific quantification validated by immunoblot is essential for unlocking the full mechanistic and therapeutic insight of BFRE in muscle–brain crosstalk research.

## 5. Conclusions

Blood flow restriction exercise appears to be a promising strategy to elevate serum BDNF levels, a key factor for neuroplasticity and cognitive health. Despite the modest number of trials currently available, the balance of evidence indicates that BFRE is able to elicit acute and, in some cases, chronic elevations in circulating BDNF that match or even surpass those produced by conventional high-load training. This effect has been replicated in healthy young adults, older women, and post-stroke patients, and appears to be mediated by the robust metabolic stress (e.g., elevated lactate) and fibrinolytic activation (increased tPA) that BFRE provokes at very low external workloads. Consequently, BFRE represents a uniquely accessible strategy for populations who cannot tolerate heavy mechanical loading yet still require potent neurotrophic stimulation to preserve or regain cognitive and motor function. However, due to the limited evidence, these findings should be interpreted with caution.

At the same time, the field is still in its infancy. Protocol heterogeneity (cuff pressure, exercise mode, session volume), small sample sizes, and reliance on single-analyte ELISAs that cannot discriminate mature BDNF from its pro-form all limit the strength of present inferences. Future multi-center RCTs should (i) standardize reporting of limb occlusion pressures relative to arterial occlusion pressure, (ii) examine sex-specific and age-related responsiveness, and (iii) adopt isoform-specific assays paired with tPA/plasmin activity measures to clarify whether BFRE consistently shifts the proBDNF:mBDNF ratio toward a neuroprotective profile. Long-term trials that link these molecular outcomes to hard clinical end-points, such as executive function, gait speed, and quality of life, will be pivotal in determining whether BFRE can move from promising laboratory intervention to routine neuro-rehabilitative and geriatric practice.

## Figures and Tables

**Figure 1 muscles-04-00019-f001:**
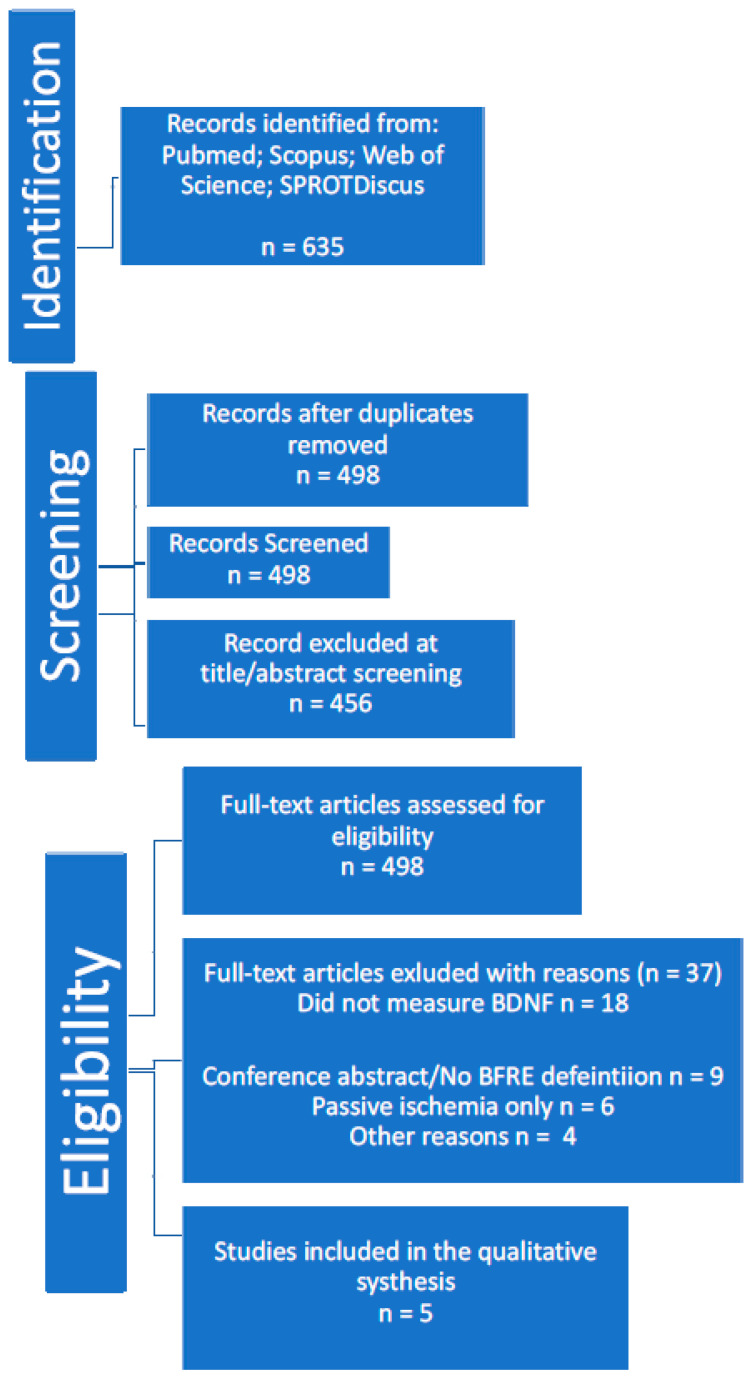
PRISMA flow diagram.

**Table 1 muscles-04-00019-t001:** Risk of bias assessment for included studies across six domains. L = low risk; S = some concerns; H = high risk; N/A = not applicable.

Study	Randomization	Allocation	Blinding	Attrition	Reporting	Other Bias
Landers et al. (2025) [14]	L	L	L	L	L	S
Du et al. (2021) [8]	L	N/A	L	L	L	S
Kargaran et al. (2021) [15]	L	L	L	L	L	S
Tsai et al. (2024) [16]	L	L	L	L	L	S
Rahmati et al. (2016) [17]	H	H	L	L	L	S

**Table 2 muscles-04-00019-t002:** Overview of included studies examining BFR exercise and BDNF.

Study (Year)	Population	Design	BFR Exercise Intervention	Comparator	BDNF Outcome
Landers et al. (2025)—in press [14]	18 healthy adults (9 F, 9 M), 34 ± 10 yrs	RCT (parallel groups)	15 min cycling (arm + leg ergometer) at ~40% VO_2_max with BFR (arm cuffs 160 mmHg; leg cuffs 300 mmHg)	Same cycling protocol without BFR	Higher in BFRE group *p* < 0.002 Acute Intervention
Du et al. (2021) [8]	24 PSD patients, 48 ± 5 yrs	Randomized crossover	Low-load resistance exercise (40% 1-RM) with BFR (cuffs 120–160 mmHg; multiple exercises, single session	(1) 40% 1-RM without BFR; (2) 80% 1-RM high-intensity exercise (each subject did all 3 conditions)	BFRE equal to HIT and higher than low-intensity *p* < 0.05 Acute Intervention
Tsai et al. (2024) [16]	66 adults, late-middle-age (~60 yrs)	RCT (3-arm)	Isometric leg resistance exercises combined with BFR (thigh cuff pressure not specified)	Group 1: resistance + WBV; Group 2: resistance + WBV + BFR; WBV-only group serves as control	No difference in BFRE group Acute Intervention
Rahmati et al. (2016) [17]	24 young men, ~21 yrs	Controlled trial (non-rand)	Cycling intervals (3 × 3 min at 50% W_max) with BFR (thigh cuffs 140–170 mmHg); 3 sessions/week for 3 weeks	Cycling without BFR; and no-exercise control group	No difference in BFRE group Acute and Chronic Interventions
Kargaran et al. (2021) [15]	24 older women, 63 ± 3 yrs	RCT (parallel groups)	Dual-task treadmill walking at 45% HRR, 20 min/session, 3×/week for 8 weeks, with BFR (thigh cuffs at 50% arterial occlusion, incremented to 70%)	Dual-task walking without BFR; plus a non-training control group	BFRE higher than control *p* < 0.005 and equal to dual task Chronic Intervention

## Data Availability

The original data presented in the study are openly available in ResearchGate at https://www.researchgate.net/profile/Josh-Landers?ev=hdr_xprf (accessed on 1 April 2025).

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
