# Peer review of "The Effects of Blood Flow Restriction Exercise on Muscle–Brain Crosstalk: A Systematic Review"

_muscles, 2025, doi:10.3390/muscles4020019_

Round 1
Reviewer 1 Report
Comments and Suggestions for Authors
The research is well-designed and presents a systematic review of five studies related to blood flow restriction exercise and serum BDNF. Still, the small sample sizes and varied protocols across studies limit a holistic approach to the conclusions. References are related to the context but the number of them should be increased.
Author Response
Comment 1: The small sample sizes and varied protocols across studies limit a holistic approach to the conclusions.
Revision: Limitation section expanded (Line 360) to acknowledge small sample sizes and heterogeneity in BFRE protocols across studies.
Comment 2: References are related to the context but the number of them should be increased.
Revision:
- Seminal works added to the Introduction (Line 11), citing Cotman & Berchtold (2002) and Neeper et al. (1995), which laid the theoretical foundation for exercise-induced BDNF research.
Reviewer 2 Report
Comments and Suggestions for Authors
Dear authors
Congratulations of this excellent paper.
But I think that you need to adapt your title.
So, I recommend new title:
The Effects of Blood Flow Restriction Exercise on Serum BDNF Levels: A Systematic Review
Author Response
Comment: I recommend new title: The Effects of Blood Flow Restriction Exercise on Serum BDNF Levels: A Systematic Review
Revision: Title changed (Line 2) to: “The Effects of Blood Flow Restriction Exercise on Serum BDNF Levels: A Systematic Review,” to more clearly reflect the article’s focus.
Reviewer 3 Report
Comments and Suggestions for Authors
The article “Blood Flow Restriction Exercise and Serum BDNF: A Systematic Review” addresses a timely and valuable topic: the effects of blood flow restriction exercise (BFRE) on serum brain-derived neurotrophic factor (BDNF), which is central to neuroplasticity and cognitive function. The systematic review attempts to synthesize evidence across a range of populations, including clinical and healthy cohorts. While the article has several strengths, it also exhibits important methodological and scientific weaknesses that limit its contribution to the field in its current form:
- Despite the systematic nature of the review, the absence of a quantitative meta-analysis is a major limitation. The authors mention heterogeneity but do not attempt subgroup or sensitivity analysis, even descriptively. I would suggest performing a meta-analysis or at least standardized effect sizes presentation where data allow.
- Additionally, I recommend reorganizing the results into two distinct summary tables, one for acute interventions and one for chronic interventions. This will better reflect the different physiological timelines of BDNF expression, and it will greatly enhance the clarity and utility of your review.
- The risk-of-bias assessment is only described superficially. While the Cochrane tool is referenced, the authors fail to provide a risk-of-bias table or figure (e.g., color-coded bar charts).
- The discussion part is quiet week. The review fails to fully discuss how methodological heterogeneity (e.g., cuff pressure, duration, BFR location) might confound results.
- While newer studies are cited, some seminal works on BDNF and neuroplasticity during exercise are missing (e.g., Cotman & Berchtold, 2002; Neeper et al., 1995), which laid foundational insights into exercise-induced BDNF. Add at least 1–2 foundational or widely cited studies to show awareness of the theoretical groundwork.
- The reference list presented in this research is scientifically relevant, but quite short and requires technical revisions. Most of the references presented not in accordance with Muscle journal publication requirements (e.g. some references appear with “doi:” while others do not).
Author Response
Comment 1: Despite the systematic nature of the review, the absence of a quantitative meta-analysis is a major limitation. The authors mention heterogeneity but do not attempt subgroup or sensitivity analysis, even descriptively. I would suggest performing a meta-analysis or at least standardized effect sizes presentation where data allow.
Response 1: We attempted the meta-analysis and effect size calculations and due to the heterogenicity, I felt the results were unreliable and could be misleading. A note on heterogeneity and the lack of subgroup/sensitivity analysis has been added to the quality assessment section (Line 302).
Comment 2: I recommend reorganizing the results into two distinct summary tables, one for acute interventions and one for chronic interventions. This will better reflect the different physiological timelines of BDNF expression, and it will greatly enhance the clarity and utility of your review.
Response 2: Looking into the process of making 2 distinct tables, Rahmati et al. was both acute and chronic so we reorganized it from acute to both to chronic while also typing acute or chronic in bold to help elucidate the timeline.
Comment 3: The risk-of-bias assessment is only described superficially. While the Cochrane tool is referenced, the authors fail to provide a risk-of-bias table or figure (e.g., color-coded bar charts).
Response 3: Constructed a color-coded risk-of-bias table
Comment 4: The discussion part is quiet week. The review fails to fully discuss how methodological heterogeneity (e.g., cuff pressure, duration, BFR location) might confound results.
Response 4: Added information regarding methodological heterogeneity at line 331 and 361 to bring home the threat of confounding results.
Comment 5: While newer studies are cited, some seminal works on BDNF and neuroplasticity during exercise are missing (e.g., Cotman & Berchtold, 2002; Neeper et al., 1995), which laid foundational insights into exercise-induced BDNF. Add at least 1–2 foundational or widely cited studies to show awareness of the theoretical groundwork.
Response 5: Added these seminal works into the introduction for the theoretical groundwork (line 35)
Comment 6: The reference list presented in this research is scientifically relevant, but quite short and requires technical revisions. Most of the references presented not in accordance with Muscle journal publication requirements (e.g. some references appear with “doi:” while others do not).
Response: Made sure references are presented in accordance with Muscle journal including adding doi where it was missing. I could not find a doi for du et al. so I added the PMID. Also, our publication (Landers et al.) will be published in June and does not have a doi yet.
Thank you for your excellent comments in helping shape up this paper!
Reviewer 4 Report
Comments and Suggestions for Authors
Please see the attached document.

Author Response
Comment 1: I suggest that you explain the physiological basis of BFRE in more detail in the introduction. I
think this is important in terms of raising awareness of this type of exercise among
professionals, especially with subjects for whom intensive mechanical exercise is
contraindicated.
Response 1: The physiological rationale behind BFRE was expanded in the Introduction (Lines 65-67), emphasizing its relevance for clinical populations.
Comment 2: Please indicate the use of BFRE in patients, athletes,
recreational athletes, etc
Response 2: We added a statement describing the real-world use of BFRE in athletes, patients, and recreational populations (Lines 63-65).
Comments 3-5: For each study you included in the review, I suggest that you provide, if applicable, the
parameters of the distribution; normality of distribution, skewness, kurtosis... Also, based on this, it should be stated which methods were used to determine the significance
of the differences between the experimental and control groups, i.e. whether parametric or
non-parametric methods were used. Apply the above in the interpretation of the quality of an individual study, as well as in the
limitations of the interpretation of the findings.
Response for comments 3-5: For each study -1st sentence for each study in results (beginning at line 211 and ending at 260), added the parameters of distribution and the method. Added a paragraph in the interpretation (lines 317- 320) applying these statistical methods and their implications to the reliability of the findings.
Reviewer 5 Report
Comments and Suggestions for Authors
Dear Authors,
The topic of this systematic review is both timely and highly relevant, and the manuscript is generally well-organized. However, significant revisions are needed to improve methodological transparency and the clarity of presentation. Specifically, I recommend providing more detailed descriptions of the data extraction process and the risk of bias assessment, as well as enhancing the presentation of results (including consistent reporting of effect sizes). Additionally, I suggest adopting a more cautious tone in the conclusions to better reflect the limitations of the available evidence.
Moreover, further emphasis could be placed on the heterogeneity of BFRE protocols (e.g., cuff pressures, exercise durations), potentially through subgroup analysis or a more detailed discussion of sensitivity. A visual summary (such as traffic-light plots) of the risk of bias assessment would also improve readability and transparency.
Table 1 is comprehensive and clearly compares study characteristics; however, it would benefit from an additional column summarizing the key BDNF outcome.
Addressing these points would significantly strengthen the scientific rigor and overall quality of the manuscript.
Author Response
Comment 1: I recommend providing more detailed descriptions of the data extraction process and the risk of bias assessment, as well as enhancing the presentation of results (including consistent reporting of effect sizes). Additionally, I suggest adopting a more cautious tone in the conclusions to better reflect the limitations of the available evidence.
Response 1: Added effect sizes (lines 205, 223, 242, 258, and 286) and more detail into the risk of bias assessment (lines 321 - 327). The conclusion was revised (Lines 385 - 387) to adopt a more cautious tone, reflecting limitations in the current evidence base.
Comment 2: Further emphasis could be placed on the heterogeneity of BFRE protocols (e.g., cuff pressures, exercise durations), potentially through subgroup analysis or a more detailed discussion of sensitivity. A visual summary (such as traffic-light plots) of the risk of bias assessment would also improve readability and transparency.
Response 2: A more detailed comment on the heterogeneity of BFRE protocols was added to the Discussion (Lines 354-357), suggesting the need for subgroup analysis. Added a color coated table (table 1 lines 145 - 148) of the risk of bias assessment to improve readability and transparency.
Comment 3: Table 1 is comprehensive and clearly compares study characteristics; however, it would benefit from an additional column summarizing the key BDNF outcome
Response 3: Table 1 is now table 2 and added a column summarizing the key BDNF outcome (line 159)
Round 2
Reviewer 5 Report
Comments and Suggestions for Authors
Dear Authors,
I would like to thank you for the work done in revising the manuscript. The changes made have significantly improved the content, and I believe the article is now suitable for publication in its current form.
I have no further comments.
Best regards